# Multi Perspectives Steganography Algorithm for Color Images on Multiple-Formats

Shahid Rahman [1], Jamal Uddin [1], Hameed Hussain [2], Salman Jan [3,*], Inayat Khan [4], Muhammad Shabir [2] and Shahrulniza Musa [3]

1  Department of Computer Science, Qurtuba University of Science and Information Technology (QUST), Peshawar 25124, Pakistan
2  Department of Computer Science, University of Buner, Buner 19290, Pakistan
3  Malaysian Institute of Information Technology, Universiti Kuala Lumpur, Kuala Lumpur 50250, Malaysia
4  Department of Computer Science, University of Engineering and Technology, Mardan 23200, Pakistan
*  Correspondence: salman.jan@unikl.edu.my

**Abstract:** The Internet and Big Data expansion have motivated the requirement for more generous stockpiling to hold and share information. Against the current era of information, guaranteeing protection and security to individuals sending data to each other is of utmost importance. The only file type that is instantly and widely used is the image. Therefore, to secure transmission, it is necessary to develop a mechanism to safeguard user data transmission. Considering this thought, it is necessary to analyze the best file type of image for essential criteria of image steganography, such as Payload, Robustness, Imperceptibility, etc., to challenge the weakness of the current algorithms. The widely used image formats are PNG, TIFF, JPEG, BMP, and GIF, which is the cause of existing methods. However, in this case, the critical softness is the credibility of the steganography, which plays a vital role in these format images to ensure the end users communicate. In this paper, a single algorithm provides several advantages for various types of images used as cover objects. However, after the critical and comparative analysis of different perspectives and some assessment metrics, the experimental results prove the importance, significance, and promising limits for these image formats by accomplishing a 4.4450% normal higher score for PSNR correlation than the next best existing methodology. Besides, in PSNR with a variable measure of code implanted in similar pictures of similar aspects, the proposed approach accomplished a 6.33% better score. Encrypting similar code sizes in pictures of various dimensions brought about a 4.23% better score. Embedding the same message size into the same dimension of different images resulted in a 3.222% better score.

**Keywords:** steganography; cover image; robustness; payload; imperceptibility

## 1. Introduction

Steganography is a mechanism of hiding messages within the cover objects such as image, audio, text, video, and network protocol so that no naked eye or attackers can detect the hidden message. Generally, steganography is the concept of a cover algorithm that tries to encrypt the secret message within some cover objects/medium transmitted over the internet that no attacker can detect. The significance of steganography is that the naked eyes or assailants cannot suspect the hidden message in any cover medium. So in this advanced era of rapid uses of technology over the internet and the uses of different image transmission, image steganography plays a vital role. Because it gives a better result that no one can suspect a noticeable change between the cover and steganographic objects [1,2]. So other cover objects, such as audio, text, and video, also use bits and can transmit a secret message using steganography. After critically analyzing the existing methods, the results prove that every method used a particular image format to hide the secret message [3]. Out of these existing methods, some methods used Portable Network Graphics (PNG) [4–7], Joint Photographic Expert Group (JPEG) [8–10], Tagged Image File Format (TIFF) [11], Red

Green Blue (RGB) [12–15] for embedding the secret message and some have tried to develop an algorithm that works for all formats steganography. But cannot obtain acceptable results or reliability between the criteria such as Payload, Transparency, Computation, Robustness, and Temper protection are shown in Figure 1, and the details of these algorithms are given in Section 2 [16].

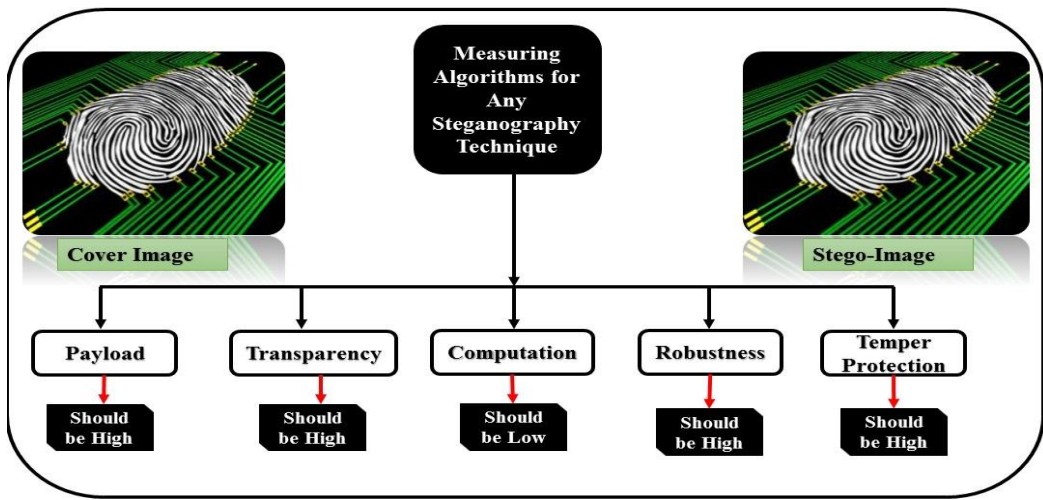

**Figure 1.** Measuring algorithm for image steganography.

Moreover, this research presents a general image steganography algorithm that works for all formats of images as a single algorithm with acceptable experimental results. So the significance of this research work is that this algorithm may utilize all image formats at a glance, giving promising results and acceptable adaption. Considering the existing methods, the proposed algorithm used as a single algorithm for embedding secret messages gives reasonable results. Let us clarify more about the motivation for the proposed algorithm used for all formats images if the researchers read this, and maybe some points that come into mind regarding how the algorithm is used for all file types images.

As we studied many existing steganographic algorithms, the point is that selecting the cover object is still a problem. Because by choosing a cover object, some methods get security but break down the steganography's other criteria. So, the problem is that the steganography algorithms require reliability between the essential criteria such as payload, temper protection, robustness, and transparency. Moreover, to increase the possibility to obtain the reliability between the criteria of steganography up to some acceptable limits, a single algorithm (Proposed) is used by just changing some parameters for the algorithm, such as Magic Matrix and Multilevel Encryption Algorithm (MLEA).

However, the proposed algorithm also provides simplicity and flexibility to the communicating bodies for selecting embedded objects. The proposed method experimentally evaluated on different images, such as texture, aerial, and RGB images, to show the proposed algorithm's significance and importance.

Furthermore, the proposed method used an image as a cover object, so an image is the collection of pixels that shows the picture element or symbolically present something. Securing the Red, Green, and Blue (RGB) channels in the range of 0–255 per block or per channel, 8 bit is used. Cover or Image steganography is embedding or covering secret information so that no one can suspect or detect that some information exists here. Providing different outputs and different results by image pixels are modified. It depends on researchers how to embed the secret information within the image pixels because everyone has their own hidden information view to achieve the reliability between the basics criteria of image steganography.

To verify the vulnerabilities or distortion of any image steganography algorithms is based on Quality Assessment metrics (QAM) such as Peak Signal to Noise Ratio (PSNR),

Means Square Error (MSE), Structural Similarity Index (SSIM), Root Means Square Error (RMSE). which are discussed later [16]. Considering the essential criteria of image steganography, achieving an acceptable stego image depends on critically analyzing the algorithms based on these QAM's and embedding secret messages to obtain more effective image steganography methods. So the content detection of all image formats, such as PNG, TIFF, JPEG, and BMP is also higher [17]. So the existing algorithms still have problems in achieving a reliable image algorithm because some have the robustness and some hold the capacity but break down other criteria of image steganography.

According to the literature, in the existing algorithms, the main problem is selecting the appropriate cover object to achieve the best steganography algorithm. The proposed algorithms have significant objectives and resisted achieving reliability between criteria such as Payload, Transparency, Temper Protection, and Computation up to some acceptable or motivational limits. Moreover, solving the problem of selecting the appropriate cover object using a single novel algorithm is discussed in the proposed methodology section below. Therefore, we analyzed and used different image formats in the proposed algorithms to tackle these issues present in current algorithms. So the proposed algorithm embeds secret information within the cover image and extracts the message from the stego image, proving the method's significance. It also gives the effective image steganography method to share images over the internet without any suspicions, violating copyrights and degrading the stego image. Now it is necessary to explain some image file formats. So there are different file formats used in the image, such as PNG, TIFF, RGB, and JPEG, which are explained one by one below.

A. Image file formats:

PNG Portable Network Graphics: This file format is widely used over the web. It is originally designed to improve and replace GIF (Graphics Interchange Format). PNG files are lossless file formats. It is a lossless image file format because its compression is without losing the image's quality. PNG can hold up to 16 million colors compared to GIF, which handles only 256 colors. It is rapidly used over the web and is also called the best image format for web images. Its unique attributes are to handle the quality of the image or to save transparency. It is not suitable for all software. It is suitable for infographics and logos [18].

Tagged Image File Format (TIFF): This format image is also used over the internet and is suitable for bitmap (BMP) because it is easily editable. It is also considered that some lose fewer image files. They do not need to change or move any information or image quality because they can change or move anything to compress the image. Tiff image files provide comprehensive image quality and file sizes. It is best used for high publications, archive copies, and quality printing. Its unique attributes are also to save transparency. It produces high files but is unsuitable for web images [19].

Bitmap Image (BMP): It is developed for windows by Microsoft. It does not involve compression and also produces quality images. Due to Microsoft's proprietary format nature, we recommend using Tiff files instead of them. The key attribute is no compression, and archive copies and scanning quality is high.

Joint Photographic Experts Groups (JPEG): It is a popular image format over the web and is widely used to present and store images over the web and cameras. JPEG is computable for most programs and devices and supports all colors. It is a loss compression to save computation time, transmission time, and storage space. It is a popular image format for digital cameras, making them ideal for non-professional print and widely used as a web image. JPEG is compressed or loses the file information and can also choose the amount of compression information while saving in image editing software such as Photoshop coral draw [20].

Due to its channels, Red, Green, and Blue (RGB) is the most widely used web format, especially for image steganography. It is also called Silicon Graphics Image (SGI) and native file graphics format. It is a color bitmap image with three channels, and every channel has 8 bits in the range of 0–255 and can make 16 million colors with the combination of these

three colors. RGB uses 24 bits per pixel depth, is widely used for image steganography, and can achieve more capacity. It does not use a palette because its colors depend on the combination of these three colors [21].

To sum up the proposed algorithm, the proposed method uses the Least significant bits based and RGB color tones for image Steganography and an indicator for which bit is inserted into a position. So the algorithm used a cover image as a collection of the pixels representing CI i, j is the location of the pixel value. SMi denotes the secret message, and SM i, j is the value location of the secret message. Considering the proposed method selecting the cover image is the fundamental problem because estimating the secret message bits and cover image bits for achieving the reliability of better quality image steganography is totally dependent. So the first cover image is transposed and then flipped into channels and divides the blue channels into four equal blocks b1-b4, which is used to embed the secret message.

Moreover, other channels mean the red channel is used for calculating the difference between secret message bits and chancel bits. So the specific adaption will present the mentioned process later in the given section, specifically used to make hurdles to the attackers. In addition, a bitmap image pixel can be considered RGB color intensity. So the proposed method used the Magic Matrix and MLEA to hide secret message bits within the image pixel bits of some Least Significant Bits (LSB) to produce better quality images. However, the experimental results prove the significance and importance of the proposed algorithm compared with previous methods available in the literature. Finally, the proposed method is the art of image steganography in two ways; the proposed method has the best adaption by selecting different image formats as a cover image and also gives us better results when compared with other existing methods. It also gives reliability between the essential criteria of image steganography up to reasonable limits.

The rest of the paper is organized in a manner; Section 2 presents a literature review, Section 3 concisely shows the proposed algorithm, Section 4 shows the main objectives, improvement, or adaptation of the proposed method, and finally concludes the proposed method in Section 5.

## 2. Related Works

For image steganography, many types of research were adapted in the recent couple of years to ensure the security and confidence of our communications between two bodies because of the advancement of today's technological world [22]. So we will discuss the review of existing methods presented in the literature related to our work. It is necessary to explain some essential factors as to why we are critically evaluating or analyzing the steganographic method instead of elaborating on the previous work. Because everyone wants confidence, Integrity, and Authenticity., their data transmission over the internet and hiding capacity is also necessary because everyone wants to share more information over the web [23]. So, Integrity means the correctness of the data, and only a permissible person can change or modify the content of the message. Authenticity means that the message must be recognized correctly and shows the actual origination of the sender. In Confidently, the secret message must be confidential, and sharing the resulting image (cover image) also must be confident that no naked eye can be detected or suspect that there is some information. Hiding Payload means the hidden capacity of the secret message within the image and can be measured in bit per pixel (bpp). Transparency or visual quality, in this, both the stego and cover images must be the same, and no distortion or noisiness is allowed. Perception means a cover and stego images must be the same without suspect by any naked eyes or visual difference. Decibels (dB) are used as a unit of PSNR calculation, and the value of the stego image more significant than 40 dB considers a better quality image (Table 1).

**Table 1.** some essential factors for evaluating any steganographic method [24–27].

| Name | Essential Factors |
|---|---|
| **MSE** | The MSE Mean Square Error and PSNR are inverse because if the MSE values are less than 1, there is no difference between the cover and stego images. Because MSE is used to find the difference between the stego and cover images |
| **NCC** | Normalize Cross Correlation (NCC) is another IQAM used to analyze how cover and stego images are related. If the value of NCC is equal to 1, both images are the same; if the value is propositional to 0, then the images are different. |
| **SSIM** | structural Similarity Index (SSIM) is used in three-part or segments Luminance, Contrast, and structural. The result of the three-part will decide the quality of both images. If all segments' values are equal to 1, then both images are the same. |
| **PSNR** | Peak Signal Noise Ratio (PSNR) is another quality assessment metric used for any image steganography method. It is calculated the quality and perception of any stego images. |
| **RMSE** | Root Mean Square Error (RMSE) in distinguishing contrast between two pictures is extremely standard since it gives an enhanced nonexclusive target examination blunder in metric utilized as a piece of numerical desires. |
| **HA** | Histogram Analysis is also an assessment concept that shows the cover and stego images histograms which can assess the changes or features of both images |
| **Stego keys** | Stego keys also use existing methods to increase security by embedding encrypted media with secret information to obtain the message correctly [24–27]. |

Figure 2 shows the abovementioned discussion as an essential factor that critically analyzes the importance and significance of any image steganographic algorithm.

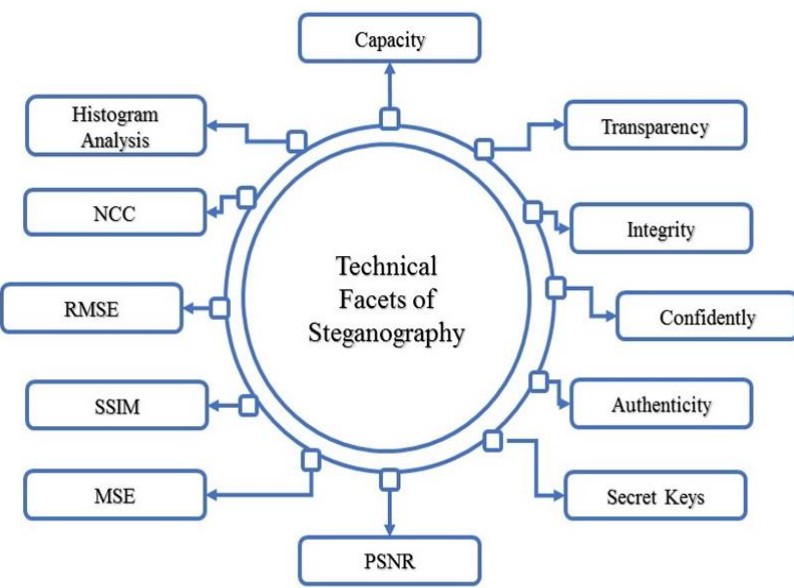

**Figure 2.** Technical facets of image steganography technique.

Figure 2 elaborates on the assessment parameters or aspects for any image steganographic algorithms. In the recent couple of years, different image steganography methods have been reported for image steganography, especially for image formats. In Table 2 the most common steganography techniques, especially PNG, Bitmap, and JPEG [11,28–48], are discussed. We critically analyzed different image steganography algorithms and elaborated on the existing methods, procedures, techniques used, cover object, and size of the embedding message. While Table 3 elaborates on the critical analysis of these different file format image steganography methods using measuring algorithms or essential criteria of image steganography. A detailed explanation is given in the tables respectively.

**Table 2.** Summary of some recent image steganography Techniques, Cover Object, and Data Used.

| Techniques Used | Uses of Data | Image Format/References |
|---|---|---|
| LSB algorithms and RC4 combination | PNG image as cover object and secret message (text) | PNG [11] |
| The k-mean algorithm is used for training the palette. Insertion is based on left-right and top-bottom | 3 different images of 512 × 512 dimensions | PNG [4] |
| Block-Entropy is used with DCT | A Grayscales image is used, which is uncompressed | JPEG [30] |
| It has used the concept of palette steganography, and insertion is only one bit per pixels based on pixel indicator | A different image is used, such as a baboon, peppers of 512 × 512 | PNG [31] |
| It is used the concept of value-based and or intensity-based insertion | It is used 640 × 480 dimensions, different images of the depth = 24 per pixel. | RGB/bitmap [32] |
| The author used high-resolution images for insertion because its transfer payload | JPEG images of different sizes and dimensions | JPEG [33] |
| In this paper, the pixel indicator method is used with LSB-modified manners | In this method, different images of 256 × 256 are used for embedding with some specific amount of secret message | Bitmap [34] |
| Matrix encoding and the canny detection method used for embedding | Different dimension 100 × 100, 80 × 80, 60 × 60 images are used for secret message embedding | RGB/bitmap [35] |
| The author used the concept of Lsb especially for android | MMS is used as a message bit and cover image is bitmap | RGB/bitmap [36] |
| In this paper, the embedding is a two-step process for increasing security by a 2D method | AES encryption techniques are used for embedding a secret message | Bitmap Image [37] |
| The author used Faye men gate for encoding and decoding the message | Reversible logic and Quantum dot cellular automata (QDCA) | Bitmap image [38] |
| This paper used SVM and Multi-level DCT | DCT is used for embedding 2 bits per pixel | RGB/bitmap [39] |
| Huffman code and AES algorithm for Gray images | 128 bits block size is used for embedding | Gray images/RGB [40] |
| In this paper, the authors used Quotient and LSB substitution to focus on high capacity | Images are divided by 3 × 3 with no overlapped sizes, two bits of LSB are used, and a quotient is applied to the remaining 6 bits | PNG and Bitmap Images [41] |
| The authors used the stego key and adaptive base image steganography concepts | Stego key is used for security, and the Multi-Level Encryption Algorithm shuffles the bits The authors also analyzed different sizes of images with different dimensions images | RGB images [42] |
| limited base mapping and Coverless Image steganography | The first extraction of the statistics between robustness mapping features. The single images are used for multiple embedding secret message | RGB images [43] |
| Using visual color intensity-based image steganography algorithm | Used two concepts, first different numbers of bits for color intensity channels. In this research, the focus is only is transparency | RGB/PNG [44] |
| Quick response-based and Histogram shifting based steganography | Used two methods quick response and shifting. This research focuses on security | RGB [45] |
| Blind Algorithm for JPEG compression attack | The paper focuses on transparency and payload with up to 8192 bits achieved | JPEG [46] |
| Reversible image steganography-based image interpolation and shifting method | First, the pixel position is changed with double scrambles operation of the image block, and logistic mapping is used to diffusion the algorithm. Based on the embedding rate of 0.67, the embedding capacity is improved up to 1,586,732 bits | PNG/RGB [47] |

**Table 3.** Critically Analysis of different formats Image steganographic Techniques based on the basics criteria for any Stego–Algorithms.

| Techniques/Uses of Data | Image Format/ References | Measuring Algorithm | | | | |
|---|---|---|---|---|---|---|
| | | Capacity | Security | Transparency | Temper Protection | Computation |
| LSB algorithms and RC4 combination. PNG image as cover object and secret message (text) | PNG | Yes | Yes | No | Yes | No |
| The k-mean algorithm is used for training the palette. Moreover, insertion is based on left-right and top-bottom. Three different images of 512 × 512 dimensions | PNG | Yes | Yes | No | No | Yes |
| Block-Entropy is used with DCT. Grayscales image is used, which is uncompressed | JPEG | No | Yes | No | Yes | No |
| It has used the concept of palette steganography, and insertion is only on bit per pixel based on pixel indicator. A different image is used, such as a baboon, peppers of 512 × 512 | PNG | Yes | Yes | No | No | Yes |
| It is used the concept of value-based and or intensity-based insertion. It uses 640 × 480 dimensions and different images of depth = 24 per pixel | RGB/bitmap | Yes | No | Yes | Yes | No |
| The author used high-resolution images for insertion because it transfers payload easily and securely, especially on Facebook. JPEG images of different sizes and dimensions | JPEG | No | Yes | No | Yes | Yes |
| This paper uses the pixel indicator method with Lsb-modified manners. In this method, different images of 256 × 256 are used for embedding with some specific amount of secret message | Bitmap | Yes | Yes | No | Yes | No |
| Matrix encoding and the canny detection method used for embedding. Different Dimension 100 × 100, 80 × 80, 60 × 60 images are used for secret message embedding | RGB/bitmap | No | Yes | No | Yes | No |
| The author used the concept of Lsb, especially for android. MMS is used as a message bit, and the cover image is a bitmap | RGB/bitmap | No | Yes | No | No | Yes |
| In this paper, embedding is a two-step process for increasing security by a 2D method. AES encryption techniques are used for embedding a secret message | Bitmap Image | No | Yes | Yes | No | Yes |
| The author used Faye men gate for encoding and decoding the message. Reversible logic and Quantum dot cellular automata (QDCA) | Bitmap image | No | Yes | Yes | No | Yes |
| This paper used SVM and Multi-level DCT. DCT is used for embedding 2 bits per pixel | RGB/bitmap | Yes | Yes | No | No | Yes |
| Huffman code and AES algorithm for Gray images. 128 bits block size is used for embedding | Gray images/RGB | No | Yes | Yes | No | No |
| In this paper, the authors used Quotient and LSB substitution to focus on high capacity. Images are divided by 3 × 3 with no overlapped sizes, two bits of LSB are used, and the quotient is applied on the remaining 6 bits | PNG and Bitmap Images | No | Yes | Yes | No | Yes |
| The authors used the stego key and adaptive base image steganography concepts. Stego key is used for security, and the Multi-Level Encryption Algorithm shuffles the bits. The authors also analyzed different sizes of images with different dimensions' images | RGB images | No | Yes | Yes | No | Yes |
| Limited base mapping and Coverless Image steganography. The first extraction of the statistics between robustness mapping features. The single images are used for multiple embedding secret message | RGB images | Yes | Yes | No | Yes | No |
| We are using visual color intensity-based image steganography algorithm. Used two concepts, first different numbers of bits for color intensity channels. In this research, the focus is only is transparency | RGB/PNG | Yes | No | Yes | No | Yes |
| Quick response-based and histogram-shifting-based steganography. Used two methods quick response and shifting. This research focuses on security | RGB | No | Yes | No | Yes | Yes |
| Blind Algorithm for JPEG compression attack focuses on transparency and payload with up to 8192 bits achieved | JPEG | Yes | No | Yes | Yes | No |
| Reversible image steganography-based image interpolation and shifting method | PNG/RGB | Yes | No | Yes | Yes | No |

In summary of the above tables, in Table 2 first, we critically analyzed variants format images based on steganography and showed their techniques, secret message embedding procedures, cover objects, hiding capacity, and their main focuses or achievements. While Table 3 elaborates on the critical analysis of these different formats of image-based steganography techniques based on the main criteria of image steganography. However, considering these different image formats used for steganography, the main points come to mind that

every format has its pros and cons, such as properties, compression, visual nature, and bit depth per pixel. So for image steganography, the question is which type of image is useful for embedding the secret message. It depends on the criteria of image steganography to achieve and other formats of the image that are rapidly used and shared over the internet. Arguably, a detailed analysis of these existing techniques shows us and gives a result of two things and considering the main problems for any image steganography; one problem is the selection of an appropriate cover object, and the other is the reliability between basics criteria or measuring algorithms for image steganography. Therefore, to tackle these issues, the proposed algorithm is designed in a manner that is used as a single algorithm procedure for selecting any format of image, especially PNG, Bitmap, and TIFF. as a cover object. The experimental results prove the reliability between the criteria of image steganography and some acceptable limit that vindicates the motivation of this research work. Selecting the cover object and embedding the secret bits within the image pixels in a manner give them tough times for the attackers. The following section elaborates on the algorithm formulation in detail.

### 3. Proposed Algorithm

In this section, we explained an abstract form of the basic mathematical formulation of the proposed algorithm and the basic definition of the main concepts used for embedding the secret message within the image. Figure 3 elaborates on the whole module of the proposed algorithm. Selecting a cover object first transposed, flipped the cover image, and separated the colors channel of the cover image. After the separation of the color channels blue channel is further divided into four equal blocks, then shuffle the sub blocks of the channel by magic matrix (discussed later) because the blue channel is used for embedding the secret message bits. On the other side, the red channel is used with secret message bits differencing calculations and then converted into the bit; after conversion into bits, applying the Multi-Level Encryption Algorithm (MLEA discussed in the given section) for making the cypher text. Finally, the cipher is embedded into shuffled blocks of blue channels. After completion of the embedding process then, rearrange the sub-images, combine the color channels, and make the intermediate stego image. The extraction algorithm is the reverse process of the embedding algorithm. The complete step-by-step process of embedding and extraction algorithms is given in Algorithms 1 and 2, respectively. Figure 3 shows the whole embedding process of the proposed algorithm.

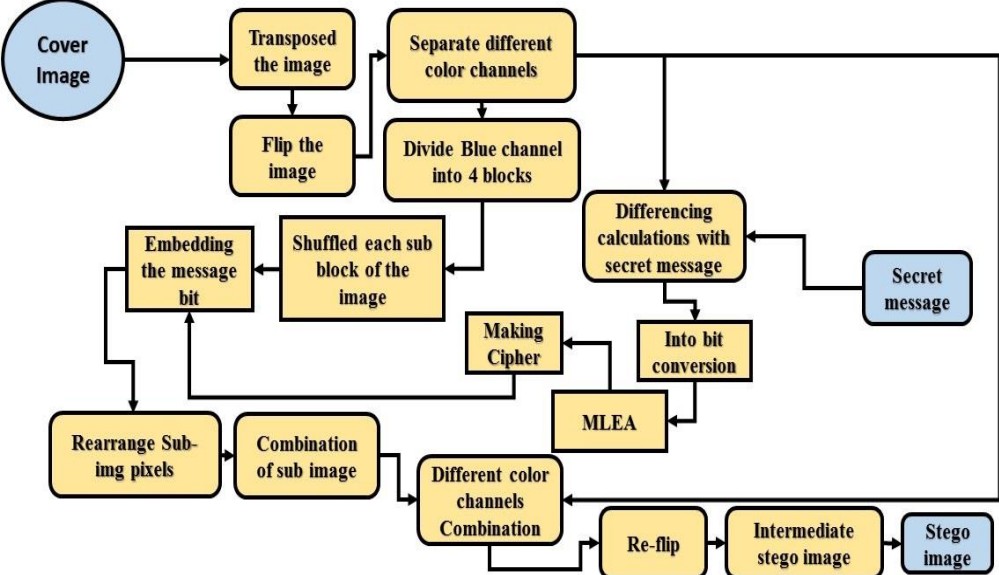

**Figure 3.** Block diagram of the proposed algorithm.

### 3.1. Magic Matrix (McMx)

Magic matrix is a Matlab function that creates a magic n-by-n matrix with an equal number of rows and columns constructed from int 1 through n2. It is necessary to create a valid matrix. Then its order must be greater than 3 and not contain any repeated numbers when taking the sum of all rows, columns, and diagonals because the summations of these three perspectives must remain the same. So McMx plays a vital role while working with image steganography because, through the magic matrix's many properties, we can shuffle the pixels of the cover image from different perspectives. For more clarity on the magic matrix, let us explain with the help of an example. Place the numbers 1 to 9 so that each row, column, and diagonal adds up to the same numbers as shown in Figure 4.

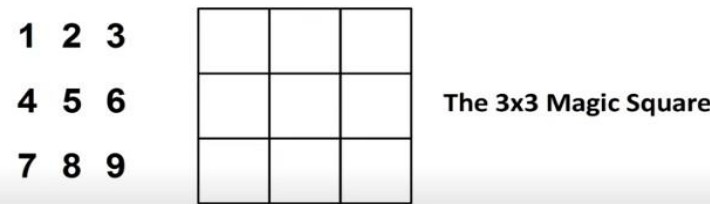

**Figure 4.** Magic Matrix (McMx) example.

Prove that there are no other ways by finding all possible arrangements. Considering the above 3 by 3 magic matrix, try to make all the possibilities; eight are shown below in Figure 5.

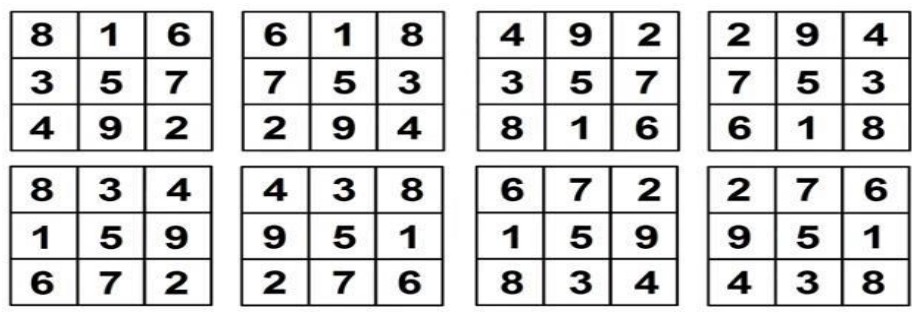

**Figure 5.** Some possibilities of the 3-by-3 Magic Matrix.

Every single one of these is a single parent, and there are reflections/rotations of one pattern. Suppose we take a copy of one of them and let us imagine the reflection of the numbers in the mirror and replace the first and third columns to provide a square of McMx shown in Figures 6 and 7.

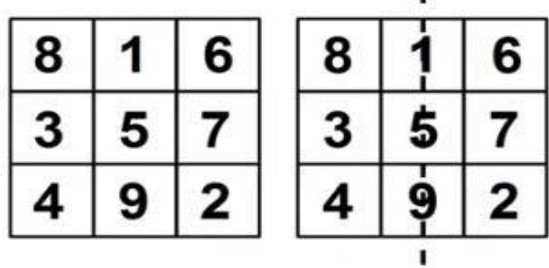

**Figure 6.** Copy of the possibility of the MM.

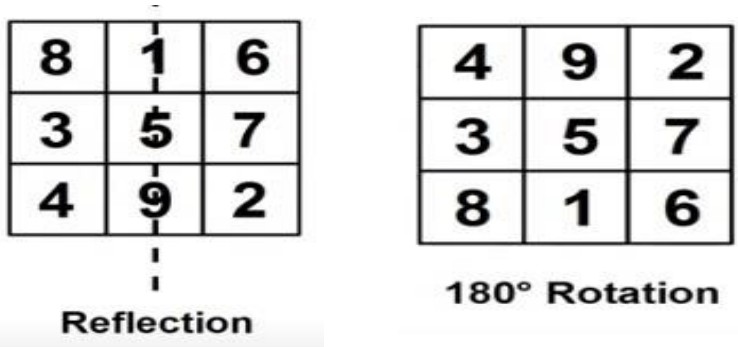

**Figure 7.** Rotation of the reflected image by 180 degrees.

If we rotate the reflected image by 180 degrees and correct the numbers of sides up, this is another magic matrix square, given in Figure 7.

Now, take a copy of the first square again and do a reflection of the cross diagonal; this will exchange all the numbers, which will generate a cross diagonal with each other shown in Figure 8.

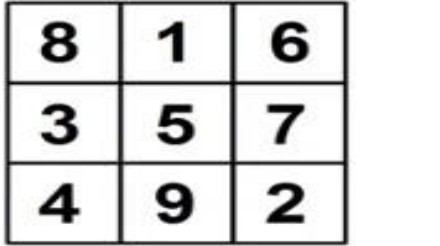 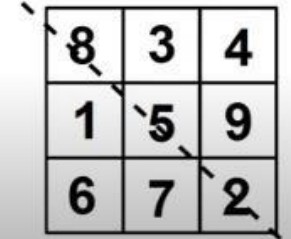

**Figure 8.** Diagonal example of MM square.

However, similarly, we can make more squares of the magic matrix by reflections, rotations and the combination of both rotations and reflections, etc.

To sum up, the Magic Matrix explains that it is the game of shuffling the numbers of the square by reflections, rotations, or a combination of both. Because its main property is how to combine or exchange the numbers in a manner that returns a matrix with no repeated numbers, and the summations of rows, columns, and diagonals remain the same. Either we can try to set the square of the matrix by taking the even number concept, odd numbers, center, and starting number of the matrix, but the main point is to obtain a matrix with an equal sum of all manipulations.

### 3.2. Multi-Level Encryption Algorithm (MLEA)

To increase the range in robustness and give the tough time for attackers, MLEA plays an essential role in image steganography, Because of their embedded functionalities for applying confidential data before embedding it to the cover image. Therefore, to obtain the benefits of these activities, the proposed method is applied to confidential data shown in Algorithm 3. The main steps of MLEA are first taking the XOR (exclusive OR) by 1. Secondly, taking the 8 bits' combination and replacing the first 4 bits with the last 4 bits and for every 8 bits' combination takes a left circular shift. Finally, divide the whole array into two blocks and, on the bases of XOR, block 1 and 2 in a way if block 1 == 1, then XOR of the block I by.

---

**Algorithm 1. Embedding Process**

---

**Input:** Cover Image CI Cx, y, Secret Information Si
**Output:** Stego Image SI

---

**1. Begin**
**2.** Transposed the cover Image (Timg) and then Flipped the image denoted by Fpimg.
**3.** Separate the color channels of the flipped image.
**4.** Now calculate the difference calculations between the red channel and secret data Sidiv
**5.** *Now convert the results of step 4 into bits of a one-dimensional array. Then applying MLEA to make the cipher text*
**6.** Now divide the channel into 4 equal blocks, which are denoted by Bc (Bc1-Bc4) and shuffles the further divided Bc channel
**7. Repeat** now select 8 bits from the secret data
**8.**      Are 1st and 2nd bits? if yes then embed into Bc1 if not the control goes to;
**9.**      Are 3rd and 4th bits? If yes, then embed it into Bc2. If ~ then control seek for another pair
**10.**      Is it the 5th and 6th bits if yes, then embed it into Bc 3. If no, then control sought from others
**11.**   Is it the 7th and 8th bits if yes, then embed it into Bc 4, and control goes to check further
**12.**      If all bits are embedded, rearrange the 4 blocks and combine all the color channels.
**13.** *Then* transposed the image and flipped the image, and made a stego–Image
**14. End**

---

**Algorithm 2. Extraction Algorithm**

---

**Input:** Stego Image S x, y
**Output:** Original Image and Secret Information Si

---

**1. Begin**
**2.**   Transposed the Stego Image (SI) and then Flipped the image denoted by Fpimg.
**3.**    Separate the color channels of the flipped image and set Flag = 1
**4.**     Now divide the BC channel into 4 equal blocks, which are denoted by (Bc1-Bc4)
**5. Repeat** now the data extracted from the said 4 blocks and check the flag setting condition
**6.**         Is flag = 1?      if yes, Extract lsb from 2 pixels of Bc1 and set Flag = 2
if not, the control goes to flag set = 2;
**7.**         Is flag = 2?      if yes, Extract lsb from 2 pixels of Bc2 and set Flag = 3
if not, the control goes to Flag set = 3;
**8.**         Is Flag = 3?      if yes, Extract lsb from 2 pixels of Bc3 and set Flag = 4
if not, the control goes to flag set = 4;
**9.**         Is flag = 4?      if yes, Extract lsb from 2 pixels of Bc4 and set Flag = 1
if not, the control goes to checking for all bits' extraction
**10.**      Are all bits extracted? If yes, then the control goes to the following control:
**11.**         Apply MLEA and converting into bits
**12.**         Calculate the difference calculations between the red channel and confidential data and combine all channels. Obtain the secret information
**13.**         If the condition of all bits becomes no, then Repeat Step 5 until all bits are extracted
**14. End**

---

**Algorithm 3. Multi-Level Encryption**

---

MLEA process on Proposed Algorithm

---

**Input:** Secret Message Si
**Output:** Concatenation of B1 and B2

---

**1. Begin**
2. Select the secret message and convert it into bits
3.   Perform bitXor (message bits, logical 1)
4.    Take the 8bits combination and replace the first 4 bits with the last 4bits
5.     Perform left circular shift to every 8bits combination
6.      Divide whole bits' array into 2 equals size blocks b1 and b2
7. After that, take a bit from b1, then check the condition
8.       Is b1 bit equal to 1?
**9.          If yes, then**
10.             Perform bitXor (b2, logical 1) and goes to b1 for next bit
**11.          If not, then**
12.             Leave the b2 bit unchanged and go to the next bit of b1
**13.       Is this the last bit?**
14.          If yes, then concatenate b1 and b2
15.          If no, then **Repeat step 7** until all bits perform
**16. End**

---

### 3.3. Mathematical Formulation of Proposed Method

The proposed method is designed in a manner; let us suppose the secret message is denoted by Si to be encrypted in cover object CI. $Ti_{mg}$ represents the transposed image, and $Fp_{img}$ shows the flipped image. $Si_{div}$ represents the differencing values with the secret

message Si and Red, Green, and Blue represent RGB channels. For the whole embedding procedure, six functions are used, which are $\alpha$, $\beta$, $\gamma$, $\delta$, $\Omega$, $\varphi$. These six functions represent the embedding algorithm's functionality in a way; the $\alpha$ is used for the cover object CI, Timg and Fpimg = $\alpha$ (CI). A accepts CI as information and returns the Fpimg (Flipped image). $\beta$ is used to divide the flipped image Fpimg into their corresponding Red, Green, and Blue channels. The blue block Bc embeds the message while the Red channel calculates the differencing value between the secret messages such as R, G, B. $\beta$ = (Fp$_{img}$). Considering the ASCII code of a letter, the third function is $\gamma$ which is used for calculating the differencing values between the secret message Si and red channel R. The function is Si$_{div}$ = $\gamma$ (Si, R). The proposed method is further enhanced using more security and for motivation, another function $\Omega$ is used, such as Si$_{div}$' = $\Omega$ (Si$_{div}$). Thus in this phase, the different values of a secret message are encrypted using MLEA. The value of ability may be increased by the proposed algorithm; the function $\delta$ is used before the embedding process using Magic Matrix (MM).

The Bc block is shuffled by $\delta$, which returns Bc' such is Bc' = $\delta$ (Bc). Now to obtain the stego image SI using the proposed embedding algorithm by a function called $\varphi$. Embedding differencing value Sidiv' in the shuffled image Bc' uses the equation SI = $\varphi$ (Si$_{div}$', Bc'). However, this is the whole mathematical formulation of the proposed algorithm; extracting the secret message on the receiver side needs the same function but in reverse order. First, use the stego image SI in this form Timg, Fp$_{img}$ = $\alpha$-1 (SI) to extract the message. The equation R, G, B = B$^{-1}$ (Fp$_{img}$) divides the flipped image into R, G, and B channels. Using this equation, Bc' = $\delta-1$ (Bc) based on $\delta-1$ this function, the Bc block is then shuffled using the Magic Matrix (MM) to obtain the shuffled Bc' block. Using this equation Si$_{div'}$ = $\varphi^{-1}$ (Bc') from Bc', we extracted the encrypted difference values Sidiv'. After that, to obtain the original differencing values' Sidiv', this equation is used Sidiv = $\Omega^{-1}$ (Si$_{div}$') using this function ($\Omega^{-1}$) and also apply MLEA in reverse order on encrypted differencing values. However, after completing the above process now, calculate the differencing values with red block R using this equation Si = $\gamma - 1$ (Si, R) to obtain the original message.

In the given section, the proposed algorithm is experimentally analyzed based on some QAM, different formats images with different existing methods in detail.

Now to sum up the proposed method, first we elaborate the notations and mathematical model and then explain the six functions used for the whole embedding and extracting algorithm presented in the proposed methodology Section 3.3. Then we explained the Magic Matrix, MLEA, which is used for embedding secret messages within the image in different manners to give a tough time to an attacker and improve the method's effectiveness and efficiency.

Then the step-by-step procedures of the embedding and extraction algorithms are explained how the message is embedded within the image and extracted.

Let's explain the whole embedding process concisely; first, we take a cover image then flipped and transpose the image. After that, convert it to three channels, Red, Green, Blue. The blue channel is used for embedding the secret message, divided into four equals block and shuffled using Magic Matrix while a red channel is used for calculating the differencing values between the red channel and secret message bits. The different values are encoded using Multi-Level Encryption Algorithm (MLEA) to make the cipher. Finally, the encoded different values in a cyclical mode (two bits per block) are embedded into the blue channel and divided into 4 equal shuffled blocks. After embedding the secret data into the blue channel, re-arrange the pixel of the sub- images of the blue channel and combine the RGB channel. After combining the RGB channel re-flip into its original form and obtain a stego image. So in the proposed method, flipping, transposing, MLEA, Magic Matrix and LSB concepts are used to embed secret messages within the image.

## 4. Experimental Results and Discussion

For a complete survey of the effectiveness of the proposed algorithm, it is critically analyzed and compared with various existing relevant methods. The result of the proposed

algorithm is given in detail in sub-sections. Against the proposed algorithm evaluating the existing research works using a dataset of many formats benchmark images downloaded from the websites of the University of Southern California-Signal and Image Processing Institute (USC-SIPI-2022) and Image Processing Place (IPP) [49,50]. These datasets contain many well-known RGB, TIFF, BMP, PNG, Aerial, Texture, and Gray Scale images of different formats of edgy and smooth colors.

This research work experimentally analyzed over 150 images of different formats and dimensions. It is also critically analyzed using different QAMs such as Peak Signal Noise Ratio (PSNR), Root Mean Square Error (RMSE), Mean Square Error (MSE), Image Fidelity (IF), Normalized Cross Correlation (NCC), Structural Similarity Index (SSIM), Quality Index (QI), Image Histogram (IH), and Correlation Coefficient (CC), which are discussed in sub-sections. However, the quantitative analysis of the proposed algorithm is investigated by the different perspectives shown in Figure 9:

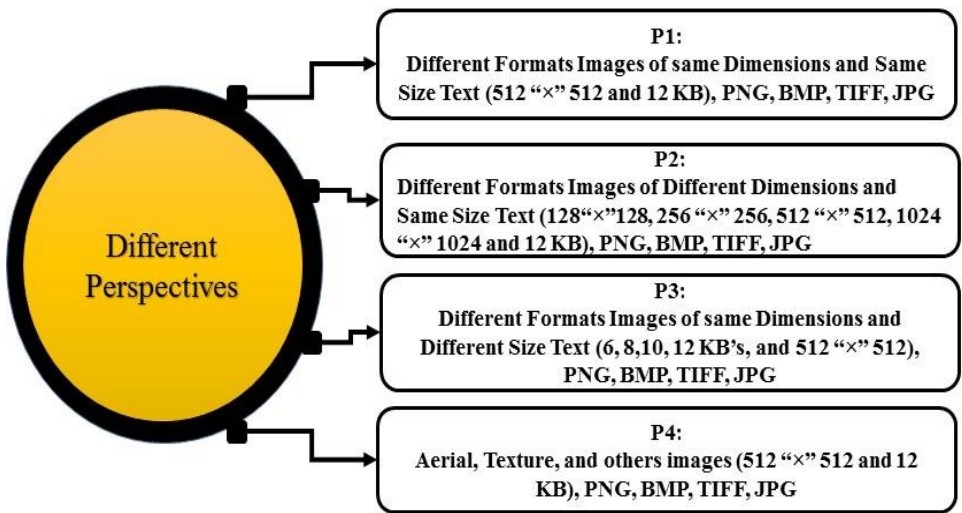

**Figure 9.** Different perspectives for proposed algorithm investigation.

Figure 9 elaborates different perspectives, for the critical analysis of the proposed work. P1, we have used these different formats (png, bmp, jpg, tiff, etc.) images with the same dimensions 512 × 512, and the size of the message 12 kb. P2 we also used these distinct formats of image with different dimensions 128 × 128, 256 × 256, 512 × 512, 1024 × 1024, and the secret message size is 12 kb. P3 present different formats of image with the same dimension, but the message size is different such as 6, 8, 10, 12 kb's. While in p4, aerial, texture, and other images of the same dimensions of the different formats and the message size is 12 kb's.

The main purpose of using these different perspectives (p1–p4) on these different formats of image (png, bmp, jpg, tiff, etc.) is to critically analyze the proposed algorithm for finding the best and appropriate procedure for which format of the image is better for which size of the message and also to identify that how many amounts of the secret message can be embedded in which dimensions of an image robustly and perceptibly to full fill the basics need of the image steganography. However, based on these 4 perspectives, the experimental results show the impacts of the distinct aspects in detail. The proposed algorithm's effectiveness to prove an experiment is conducted with over 150 experimental images and well-known pictures such as Mandrill (Baboon), Lake, Splash, Girl, Living Room, Tree, House, Jet Plane, and Peppers. Which are shown in Figures 10 and 11.

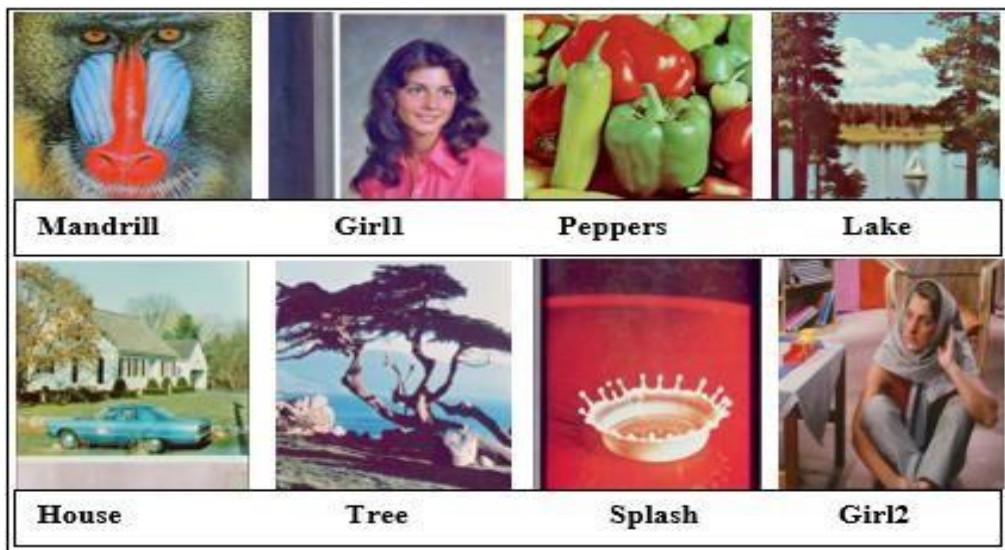

**Figure 10.** Experimental Images for the proposed algorithm.

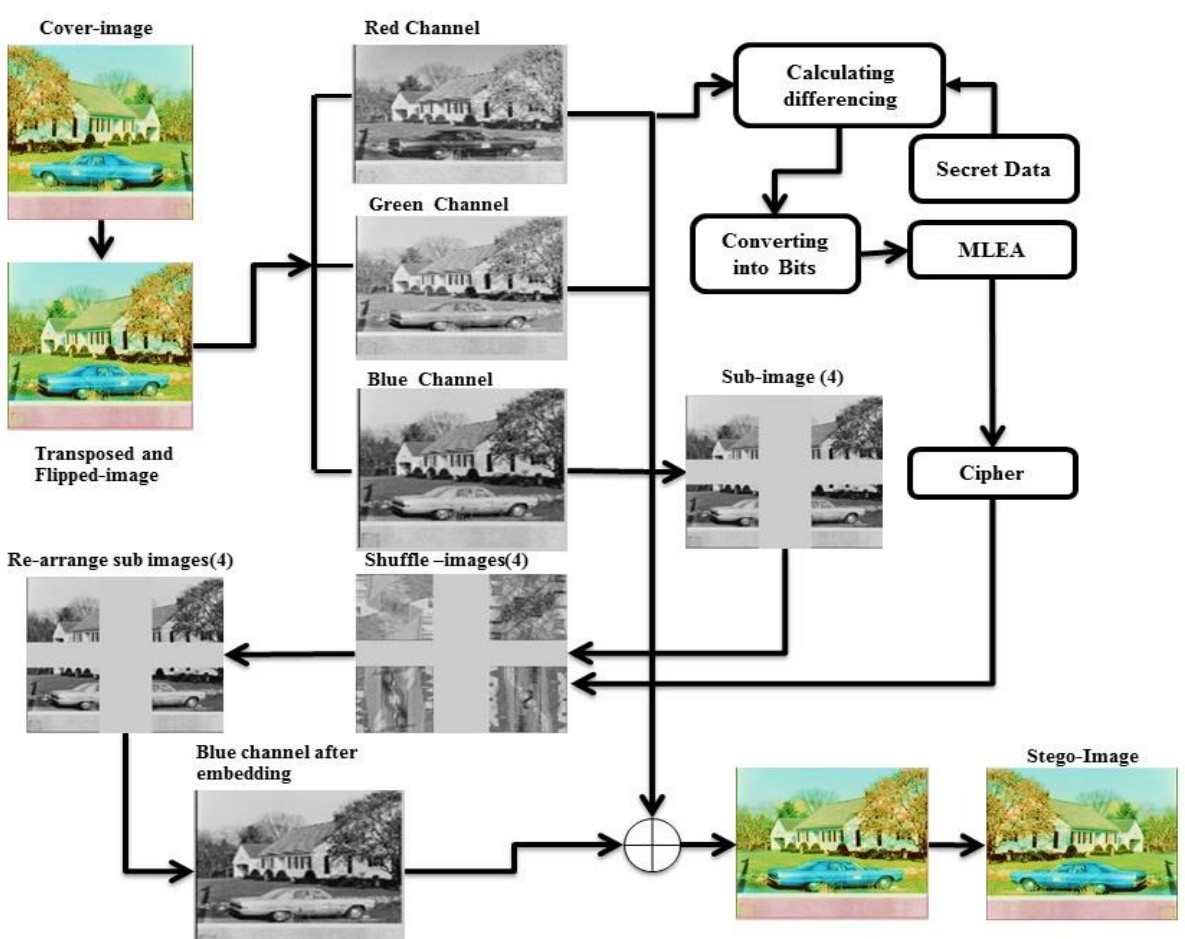

**Figure 11.** Resulting overview of the proposed method in one image.

### 4.1. Perspective One P1

Considering the perspective P1 based on these experimental images of different formats such as PNG, TIFF, BMP, and JPG, and the encoded message size is the same for all formats images, which is 12 KB, the proposed algorithm is critically analyzed, and the result shows the importance of the algorithm based on one of the QAM is PSNR shown

in Table 4. PSNR: Peak Signal Noise Ratio is a fundamental parameter for determining the quality of both the stego and cover images. If the value of PSNR is greater than 30 dB, consider it a quality image. The formula is given in Equation (1).

$$PSNR = 10\log_{10}\left(\frac{C_{max^2}}{MSE}\right) \tag{1}$$

**Table 4.** Results of PSNR values of the proposed algorithm on different formats image with exact size text.

| Images | Message Size | PSNR Values of the Proposed Algorithm | | | |
|---|---|---|---|---|---|
| | | PNG | TIFF | BMP | JPG |
| Mandrill | 12 KB | 84.321 | 83.123 | 82.212 | 85.220 |
| Girl1 | 12 KB | 90.223 | 87.765 | 83.121 | 88.980 |
| Peppers | 12 KB | 75.890 | 70.654 | 70.876 | 75.987 |
| Lake | 12 KB | 87.098 | 79.986 | 82.097 | 84.643 |
| House | 12 KB | 90.854 | 89.654 | 87.001 | 87.087 |
| Tree | 12 KB | 88.001 | 87.009 | 87.091 | 87.098 |
| Splash | 12 KB | 86.065 | 87.120 | 83.087 | 86.076 |
| Girl2 | 12 KB | 80.008 | 81.987 | 84.986 | 86.121 |
| Average of 150 images | | 85.632 | 85.162 | 84.689 | 85.972 |

### 4.2. Perspective Two P2

Based on perspective P2, different dimensions' images of different formats with the exact size text 12 KB, the proposed method was critically analyzed, and the result shows the significance of the method shown in Table 5.

**Table 5.** Results of PSNR values of the proposed algorithm based on Perspective 2.

| Images | PSNR Values of Proposed Algorithm (128 × 128, 256 × 256, 512 × 512, 1024 × 1024/12 KB) | | | | | | | | | | | | | | | |
|---|---|---|---|---|---|---|---|---|---|---|---|---|---|---|---|---|
| | PNG | | | | TIFF | | | | BMP | | | | JPG | | | |
| | 128 | 512 | 256 | 1024 | 128 | 512 | 256 | 1024 | 128 | 512 | 256 | 1024 | 128 | 512 | 256 | 1024 |
| **Mandrill** | 77.32 | 84.32 | 84.32 | 84.32 | 86.12 | 83.32 | 83.32 | 81.32 | 84.21 | 80.32 | 81.32 | 70.32 | 85.22 | 89.32 | 87.32 | **84.32** |
| **Girl1** | 76.32 | 88.21 | 87.22 | 89.23 | 79.31 | 81.11 | 77.09 | 82.21 | 73.12 | 81.32 | 80.32 | 74.31 | 87.32 | 83.32 | 87.32 | **81.11** |
| **Peppers** | 74.21 | 87.31 | 84.21 | 80.91 | 74.32 | 83.22 | 82.31 | 83.32 | 74.12 | 77.22 | 71.22 | 79.23 | 86.32 | 87.21 | 77.22 | **85.32** |
| **Lake** | 72.12 | 86.21 | 87.01 | 79.99 | 73.32 | 86 | 85.11 | 84.23 | 72.11 | 70.21 | 71.21 | 70.9 | 74.21 | 88.21 | 88.21 | **86.33** |
| **House** | 75.21 | 86.21 | 82.23 | 88.11 | 73.21 | 85.33 | 86 | 79.91 | 72.22 | 79.01 | 77.01 | 79.9 | 82.12 | 86.01 | 89.01 | **85.19** |
| **Tree** | 80.23 | 86.01 | 86.01 | 87.99 | 71.11 | 84 | 85.01 | 76.99 | 75.31 | 81.23 | 76.23 | 78.13 | 75.21 | 87.11 | 86.23 | **84.12** |
| **Splash** | 89.09 | 85.21 | 83.09 | 87.99 | 75.21 | 85.21 | 85.22 | 85.11 | 79.43 | 81.11 | 76 | 77.9 | 80.23 | 88 | 86.31 | **85.32** |
| **Girl2** | 86.23 | 85.11 | 82.21 | 88.09 | 79.23 | 83 | 83 | 87.99 | 72.39 | 82 | 76.09 | 75.09 | 89.09 | 85.09 | 76.09 | **86.31** |
| **Av.of 150 images** | 78.84 | 86.07 | 84.54 | 85.83 | 76.48 | 83.90 | 83.38 | 82.64 | 75.36 | 79.05 | 76.18 | 75.72 | 82.47 | 86.78 | 84.71 | 84.75 |

### 4.3. Perspective Three P3

The proposed algorithm is significantly analyzed using perspective P3, which formats images of the exact dimensions with different sizes of the secret message shown in Table 6.

### 4.4. Perspective Four P4

Now, using perspective P4, aerial and texture images of different formats of the same size with a 12 Kb size message. Figure 12 shows some aerial and texture images, namely, Aerial 1-3 and txtr 1-3. The experimental results are shown in Table 7.

**Table 6.** Results of PSNR values of the proposed algorithm based on Perspective 3.

| Images | PSNR Values of Proposed Algorithm (6, 8, 10, and 12 KB's/512 × 512 Image) | | | | | | | | | | | | | | | |
|---|---|---|---|---|---|---|---|---|---|---|---|---|---|---|---|---|
| | PNG | | | | TIFF | | | | BMP | | | | JPG | | | |
| | 6 KB | 8 KB | 10 KB | 12 KB | 6 KB | 8 KB | 10 KB | 12 KB | 6 KB | 8 KB | 10 KB | 12 KB | 6 KB | 8 KB | 10 KB | 12 KB |
| Mandrill | 87.32 | 84.32 | 83.32 | 83.32 | 86.12 | 83.32 | 83.32 | 81.32 | 84.21 | 80.32 | 81.32 | 70.32 | 88.32 | 83.44 | 82.32 | 82.32 |
| Girl1 | 86.32 | 82.21 | 82.22 | 80.23 | 79.31 | 81.11 | 77.09 | 82.21 | 73.12 | 81.32 | 80.32 | 76.31 | 85.32 | 82.24 | 82.22 | 80 |
| Peppers | 84.11 | 84.33 | 83.21 | 80.91 | 74.32 | 83.22 | 82.31 | 83.32 | 74.12 | 77.22 | 71.22 | 79.23 | 88.21 | 85.44 | 83.21 | 80.91 |
| Lake | 79.01 | 83.88 | 83.01 | 77.99 | 73.32 | 86 | 85.11 | 74.23 | 72.11 | 70.21 | 71.21 | 80.9 | 79.01 | 83.88 | 83.01 | 79.99 |
| House | 85.01 | 86.21 | 81.23 | 80.11 | 73.21 | 85.33 | 86 | 79.91 | 72.22 | 79.01 | 77.01 | 79.9 | 85.02 | 86.21 | 81.23 | 80 |
| Tree | 80.23 | 86.98 | 83.01 | 81.99 | 71.11 | 84 | 80.01 | 76.99 | 75.31 | 81.23 | 76.23 | 78.13 | 80.23 | 86.98 | 83.01 | 81.99 |
| Splash | 89.98 | 85.21 | 82.09 | 80.99 | 75.21 | 85.21 | 79.22 | 75.11 | 79.43 | 81.11 | 76 | 77.9 | 89.78 | 85.12 | 84 | 81.99 |
| Girl2 | 86.13 | 85.09 | 82.21 | 80.09 | 79.23 | 83 | 79 | 79.99 | 72.39 | 82 | 76.09 | 75.09 | 86.13 | 85.09 | 82.21 | 80 |
| Av of 150 images | 84.76 | 84.78 | 82.54 | 80.70 | 76.48 | 83.90 | 81.51 | 79.14 | 75.36 | 79.05 | 76.18 | 77.22 | 85.25 | 84.80 | 82.65 | 80.90 |

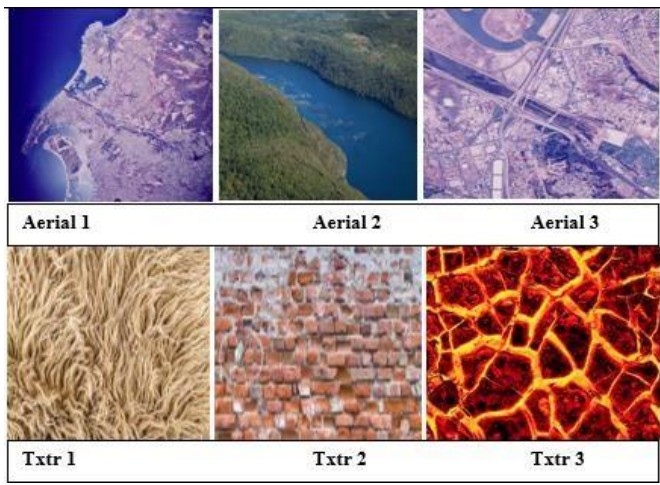

**Figure 12.** Aerial and Texture Image dataset.

**Table 7.** Results of PSNR values of Proposed Algorithm based on Aerial and Texture Images of different formats P4.

| Images | Message Size | PSNR Values of the Proposed Algorithm Based on Aerial and Texture Images of Different Formats (512 × 512) | | | |
|---|---|---|---|---|---|
| | | PNG | TIFF | BMP | JPG |
| Aerial 1 | 12 KB | 84.321 | 76.113 | 78.212 | 84.22 |
| Aerial 2 | 12 KB | 88.223 | 77.765 | 63.121 | 85.98 |
| Aerial 3 | 12 KB | 85.89 | 79.654 | 73.876 | 78.987 |
| Txtr 1 | 12 KB | 79.098 | 69.986 | 77.097 | 75.643 |
| Txtr 2 | 12 KB | 80.854 | 79.654 | 87.001 | 86.087 |
| Txtr 3 | 12 KB | 78.001 | 87.009 | 87.091 | 87.098 |
| Average of 150 images | | 82.73 | 78.36 | 77.73 | 83.00 |

*4.5. Comparison of the Proposed Algorithm with Other Recently Reported Research Work Based on PSNR*

In Table 8 the comparative analysis of the proposed algorithm with other recent research works based on their resulted in PSNR values is presented. After analyzing the research work with other existing methods based on PSNR, the results show that it outperforms others. Moreover, the experimental results take place over 150 images of different image formats in which the average PSNR value of the proposed algorithm results is 81.70, which shows the significance and motivation of the proposed research work.

**Table 8.** PSNR Results of the proposed method with other research works.

| Image | Average Results of PSNR, Compared Proposed Algorithm with Existing Research Works Using Different Image Formats | | | | | | | |
|---|---|---|---|---|---|---|---|---|
| | AbdelRaouf et al. [43] | Luxi et al. [42] | Naz et al. [33] | Peter et al. [44] | Mehta et. al. [45] | Ye, H et. al. [46] | Arsyia et al. [49] | Proposed Algorithm |
| Mandrill | 68.342 | 78.043 | 79.003 | 80.324 | 83.987 | 81.098 | 84.321 | 85.001 |
| Girl1 | 69.098 | 68.321 | 69.432 | 79.221 | 69.662 | 76.321 | 79.978 | 80.901 |
| Peppers | 70.321 | 71.211 | 68.991 | 69.001 | 70.002 | 74.321 | 72.212 | 78.001 |
| Lake | 75.009 | 73.002 | 75.221 | 75.001 | 76.779 | 79.001 | 80.221 | 81.001 |
| House | 67.087 | 67.900 | 70.009 | 71.988 | 69.900 | 77.009 | 81.007 | 81.999 |
| Tree | 79.876 | 79.991 | 80.002 | 81.008 | 80.321 | 79.987 | 82.009 | 83.001 |
| Splash | 77.002 | 76.992 | 79.900 | 80.876 | 81.098 | 80.876 | 81.987 | 82.002 |
| Average | 72.39 | 73.64 | 75.79 | 76.77 | 75.96 | 79.09 | 80.25 | 81.70 |

### 4.6. Results of Proposed Algorithm Based on Cumulative IQAM's

Table 9 presents the result of different Quality Assessment Metrics such as SSIM, MSE, RMSE, and NCC. RMSE: Using Root Mean Square Error (RMSE) in distinguishing contrast between two pictures is extremely normal since it gives an enhanced nonexclusive target examination blunder in metric utilized as a piece of numerical desires. MSE: Mean Square Error compares the original and stego image. Both stego and cover images are said to be equal if the MSE equals zero. So MSE should be the least possible. NCC: Normalize Cross Correlation is another IQAM used to analyze how cover and stego images are related. If the value of NCC is equal to 1, both images are the same; if the value is propositional to 0, then the images are different. SSIM: Structural similarity index is used three-part or segment Luminance, Contrast and structural. The result of the three-part will decide the quality of both images. If all segments' values are equal to 1, then both images are the same. Formulas for each in the given Equation (2):

$$
\text{MSE} = \text{MSE} = \frac{1}{MN}\sum_{x=1}^{M}\sum_{y=1}^{N}\left(S_{xy} - C_{xy}\right) \qquad \text{RMSE} = \sqrt{\left(\frac{1}{N}\right)\sum_{x=1}^{N}(C_x - S_x)^2}
$$
$$
\text{NCC} = \frac{\sum_{x=1}^{M}\sum_{y=1}^{N}(S(x,y)*C(x,y))}{\sum_{x=1}^{M}\sum_{y=1}^{N}S(x,y)^2} \qquad \text{SSIM}(X,Y) = \frac{\left(2\mu_x\mu_y+C_1\right)\left(2\sigma_{xy}+C_2\right)}{\left(\mu_x{}^2+\mu_y{}^2+C_1\right)\left(\sigma_x{}^2+\sigma_y{}^2+C_2\right)}
\tag{2}
$$

**Table 9.** Results of Proposed Algorithm using Cumulative Image Quality Assessment Metrics.

| IQAM | Name, Dimension, and Secret Message Size. $512 \times 512$ and 12 KB | | | | | | |
|---|---|---|---|---|---|---|---|
| | Mandrill | Girl1 | Peppers | Lake | House | Tree | Splash |
| SSIM | 0.999 | 1 | 1 | 0.999 | 1 | 0.999 | 0.999 |
| MSE | 0.021 | 0.111 | 0.001 | 0.101 | 0.122 | 0.011 | 0.121 |
| NCC | 0.989 | 0.999 | 0.999 | 0.998 | 0.999 | 0.899 | 1 |
| RMSE | 0.025 | 0.011 | 0.125 | 0.022 | 0.011 | 0.021 | 0.111 |

After analyzing the proposed algorithm with these parameters, the results prove the importance and significance of the research work. Experimental results take place over 150 images of different formats of the same dimension, $512 \times 512$, with the exact size of the secret message 12 KB in which the results show the values of MSE = 0.007, RMSE = 0.096, SSIM = 0.998, and NCC = 0.995, which shows that the stego and the cover image is identical.

### 4.7. Security Analysis for the Proposed Algorithm

Table 10 presents the detailed results of the proposed algorithm critically analyzed based on some security analysis parameters used for image steganography, which are correlation Coefficient (CC), Quality Index (QI), and Image Fidelity (IF). Correlation Coefficient: CC play a vital role due to its properties [51]. To find the extent and direction of the linear correlation of 2 random variables. If the CC values of two random variables are close to 1 mean, both are closely related, while the values become 0, showing the exact

changes between them. CC can be defined using this formula. Quality Index: The quality of the stego image can be measured using a Quality Index. QI can be found using the given formula. Where n is the number of pixels in the image, T is the Stego image, and H presents the cover image. When the resulting values are 1 of both stego and cover images, it presents the quality image; if the values of both images are −1, then it shows the dissimilarity between them. The formula for cover and stego image quality can be found in the range of −1 and 1, denoted by Q. Image Fidelity: To show the security of any steganographic algorithm, Image Fidelity (IF) is another metric used IF can be calculated by the given formula for each in Equation (3). After analyzing this research work based on these parameters, the results show outperforms.

**Table 10.** Results of the Proposed Algorithm using different Security Analysis Metrics.

| | Security Analysis of the Proposed Algorithm using Different Formats Image (512 × 512/16 KB Message Size) | | | | | | | | | | | |
| Images | Correlation Coefficient | | | | Quality Index | | | | Fidelity | | | |
| | PNG | BMP | TIFF | JPG | PNG | BMP | TIFF | JPG | PNG | BMP | TIFF | JPG |
|---|---|---|---|---|---|---|---|---|---|---|---|---|
| Mandrill | 0.999 | 0.897 | 0.876 | 0.999 | 1 | 0.866 | 0.999 | 1 | 0.999 | 0.897 | 0.988 | 0.999 |
| Girl1 | 0.998 | 0.797 | 0.855 | 0.991 | 0.999 | 0.877 | 0.876 | 0.999 | 1 | 0.897 | 0.888 | 1 |
| Peppers | 1 | 0.887 | 0.879 | 1 | 0.999 | 0.897 | 0.899 | 1 | 1 | 0.788 | 0.844 | 0.997 |
| Lake | 0.999 | 0.889 | 0.899 | 0.999 | 1 | 0.899 | 0.899 | 1 | 0.997 | 0.876 | 0.754 | 1 |
| House | 1 | 0.955 | 0.999 | 0.999 | 1 | 0.874 | 0.789 | 0.998 | 1 | 0.866 | 0.876 | 0.999 |
| Tree | 1 | 0.888 | 0.898 | 0.997 | 0.999 | 0.897 | 0.888 | 1 | 1 | 0.988 | 0.999 | 1 |
| Splash | 1 | 0.897 | 0.888 | 1 | 1 | 0.877 | 0.887 | 1 | 0.998 | 0.877 | 0.966 | 0.998 |
| Average | 0.999 | 0.887 | 0.899 | 0.998 | 1.000 | 0.884 | 0.891 | 1.000 | 0.999 | 0.884 | 0.902 | 0.999 |

$$
\begin{aligned}
&\text{Correlation Coefficient CC}\\
&I = \frac{\sum_i (x_i - x_m)(y_i - y_m)}{\sum_i \sqrt{\sum_i (x_i - x_m)^2}\sqrt{\sum_i (y_i - y_m)^2}}\\
&\text{Image Fidelity IF}\\
&\boldsymbol{IF} = 1 - \frac{\sum_{i,j}(P(i,j) - S(i,j))^2}{\sum_{i,j}(P(i,j) \times S(i,j))}\\
&\text{Quality Index QI}\\
&\boldsymbol{Q} = \frac{4\sigma_{HT}H'T'}{(\sigma_H^2 + \sigma_T^2)(H'^2 + T'^2)}\\
&\sigma_H^2 = \frac{1}{N-1}\sum_{i=1}^{N}(H_i - H')^2\\
&H' = \frac{1}{N}\sum_{i=1}^{N}H_i - T = \frac{1}{N}\sum_{i=1}^{N}T_i\\
&\sigma_H^2 = \frac{1}{N-1}\sum_{i=1}^{N}(T_i - T')^2
\end{aligned}
\tag{3}
$$

*4.8. Histogram Analysis*

Figure 13 shows the histograms of some images, namely, 'a' for Mandrill, 'b' for Splash, and 'c' for Lake, which was critically analyzed using the proposed algorithm. We can find the exact occurrences of each image pixel using a Histogram. When the secret message is embedded in the cover image, the high similarity between the stego and cover image can be found using a Histogram because it shows the bit of distortion between the stego and cover image after embedding the secret message. However, the proposed algorithm is also critically investigated for cover and stego images of different formats using a Histogram of over 150 images. So the results outperform the others. CH shows the cover image histogram, and SH shows Stego image Histogram.

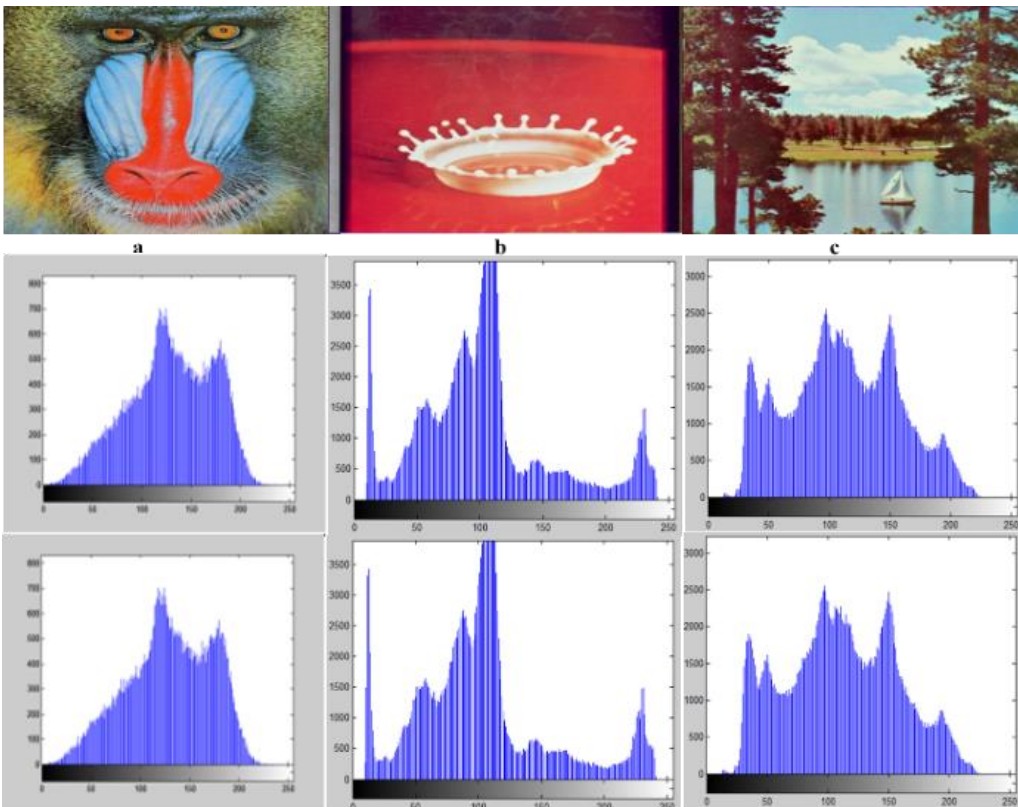

**Figure 13.** Image Histograms of both Cover and Stego images of the proposed algorithm of some images.

## 5. Conclusions

In this study, the proposed algorithm used different formats of an image used as cover objects, MLEA, Magic matrix. The proposed algorithm depends on the abstract explorations of the different image channels to find which image type is better and appropriate for embedding the secret message. Furthermore, to hide a large amount of secret information within the cover media, we divided one of the cover image channels into four equal blocks. For security, transparency, computation, and temper protection, the proposed algorithm used different concepts such as Magic Matrix, MLEA, Flipping, and Transposition function to obtain a high-quality stego image. The proposed research work is also further enhanced by different analysis viewpoints. Considering different perspectives and different QAMs such as (Peak Signal Noise Ratio (PSNR), Root Mean Square Error (RMSE), Mean Square Error (MSE), Image Fidelity (IF), Normalized Cross Correlation (NCC), Structural Similarity Index (SSIM), Quality Index (QI), Image Histogram (IH), and Correlation Coefficient (CC)) the proposed algorithm experimentally assessed with others existing research works. The proposed algorithm attains at least a 28% improvement, and the experimental results prove that the outcome is highly supportive. Due to its nature, the proposed algorithm provides many benefits, such as sending different secret information using different formats of image to avoid form steganalysis in the time of need to share data securable and adaptively. So, future work must explore cover steganography with Machine learning and Deep Learning concepts to propose best embedding procedures or architecture for selecting appropriate cover objects for concealing secret messages. Moreover, a better system should be able to hide data within the cover media to detect which cover object is suitable for which type of secret information to fully fill the need for secure communication and make a reliable system.

**Author Contributions:** Conceptualization, J.U.; Methodology, H.H. and S.M.; Software, H.H.; Validation, S.M.; Formal analysis, H.H. and I.K.; Investigation, I.K.; Resources, I.K. and S.M.; Data curation, M.S.; Writing—original draft, S.R.; Writing—review & editing, S.R.; Visualization, M.S.; Supervision, J.U.; Project administration, S.J.; Funding acquisition, S.J. All authors have read and agreed to the published version of the manuscript.

**Funding:** This research is funded by the Malaysian Institute of Information Technology, Universiti Kuala Lumpur.

**Institutional Review Board Statement:** Not applicable.

**Informed Consent Statement:** Informed consent was obtained from all subjects involved in the study.

**Data Availability Statement:** The data presented in this study are available on request from the authors.

**Acknowledgments:** This manuscript is supported by the Malaysian Institute of Information Technology, Universiti Kuala Lumpur.

**Conflicts of Interest:** The authors have no conflict of interest regarding the publication of this paper.

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
