# Peer review of "Multi Perspectives Steganography Algorithm for Color Images on Multiple-Formats"

_sustainability, doi:10.3390/su15054252_

Round 1

Reviewer 1 Report

1. Introduction is too general and broad. In addition, there are also things that are confusing.

2. In the second page of the manuscript, lines 36 to 41, the logic is confusing, would be better redescribed.

3. The purpose of this research is not clear, payload/capacity? security/undetectability? robustness? imperceptibility? other?

4. Table 1 is also confusing.

5. The proposed method is slightly unclear about its contribution, it only combines the LSB and MLEA methods. would be better to improve.

6. Writing no image using brackets [] makes confusion, is this an image number or a reference.

7. The comparisons and tests presented in Table 4 need to be explained how you do the comparisons. 

8. It is not explained the maximum payload size used. Presentation of results using a non-standard payload size, should use the BPP size.

9. The measuring instruments used are PSNR, MSE, RMSE, SSIM and NCC. It is only used for imperceptibility measurement only. Not really explained how to measure it. Then for NCC, is it used to calculate the quality of the stego image or the extraction result?

10. It can be seen that the PSNR measurement results are very good, but this value is very strange. How is it possible for an image with a larger dimension to produce a worse PSNR value? In fact, if the image/cover is bigger, the PSNR will be better because the message size is not significant compared to the image size.

11. site this relevant paper 

Sadad, Tariq, et al. "A Review on Multi-organ Cancer Detection Using Advanced Machine Learning Techniques." Current medical imaging 17.6 (2021): 686-694.

Author Response

Reviewer#1, Concern#1: Introduction is too general and broad. In addition, there are also things that are confusing.

Author response: First of all, thanks for the reviewer's comment. We feel the revised manuscript has improved by addressing the reviewer's comments. We carefully revised the manuscript to add and make it abstract and desired introduction.

Author action: We updated the manuscript accordingly to the valuable comments of the reviewer, and also rewrote the introduction section.

Reviewer#1, Concern#2:  In the second page of the manuscript, lines 36 to 41, the logic is confusing, would be better redescribed.

Author response: Thank you for the keen suggestion for clarity and removing ambiguity. We revised and redescribed accordingly.

Author action: In the revised manuscript, we updated accordingly (has been highlighted).

Reviewer#1, Concern#3: The purpose of this research is not clear, payload/capacity? security/undetectability? robustness? imperceptibility? other?

Author response: We are sorry, and thank you for your comments. The main purpose of this research is to achieve reliability between the basic criteria of image steganography for making a reliable system up to some extent because existing methods have better results but only on one or two parameters but broke down the others. Today, the most used for transmission is one file type, which is an image. Therefore, the proposed method is critically analyzed based on distinct perspectives and different formats to fulfil the basic criteria of image steganography up to some acceptable limit.

Author action: We updated the whole manuscript by rewriting the corresponding description to address the valuable comments on page 2 and 3.

Reviewer#1, Concern#4: Table 1 is also confusing.

Author response:  We are sorry for the insufficient discussion and Thanks for your comment. Basically, Table 1 presents some recent image steganography Techniques, Cover Objects, and Data Used. Reviewer 2 also suggested that paragraph lines 210-223 would be better in table format, so now the comments about Table 1 will be considered Table 2.

Author action: We updated the manuscript by adding a concise explanation at the end of Tables 1-3.

Reviewer#1, Concern#5: The proposed method slightly unclear about its contribution, it only combines the LSB and MLEA methods. would be better to improve.

Author response:  Thanks for the comments of the reviewer. We used Magic Matrix, MLEA, which are used for embedding secret messages within the image differently to give the attacker a tough time and improve the method's effectiveness and efficiency. The step-by-step procedures of the embedding and extraction algorithms are explained, and how the message is embedded within the image and extracted.

Author action:  We updated and revised the whole proposed methodology of the manuscript to make it clear and removed the ambiguity by adding the desired explanation before embedding and extraction algorithm section.

Reviewer#1, Concern#6: Writing no image using brackets [] makes confusion, is this an image number or a reference.

Author response: Thank you for a good suggestion. It is a reference, not an image number.

Author action:  We modified all figures' captions and removed image references. (highlighted).

Reviewer#1, Concern#7: The comparisons and tests presented in Table 4 need to be explained how you do the comparisons.

Author response: We are sorry that we did not clearly state how we obtained the experimental results in Table 4. In Table 4 we critically analyzed the proposed methods using different dimensions (128x128, 256x256, 512x512, 1024x1024 with message size 12 kb’s) and different images and they resulting PSNR values. However, the comparative analysis of the proposed method is presented in Table 8. For a fair comparison, first, we collected the resulted in percent values (PSNR) of the existing methods from the papers and verified them.

Author action: We updated the manuscript by adding more explanation about the experimental result for Table 4-8. (has been highlighted).

Reviewer#1, Concern#8: It is not explained the maximum payload size used. Presentation of results using a non-standard payload size, should use the BPP size.

Author response: Thanks for the comments, and sorry for confusing you due to the lack of sufficient explanation. Actually, in the embedding algorithm, after calculating the differencing values between the red channel and secret data, converting into bits of 1D array and applying MLEA to make a cipher.  So 4 equals divided blocks of the blue channel is used for every 1st two bits into block 1, and every 3rd and 4th bit into 2nd block respectively.

Author action: We updated the manuscript and added the descriptive explanation of the embedding process to address the valuable comments of the reviewer. Tables 4-6 elaborate the embedding perspectives highlighted in the revised manuscript.

Reviewer#1, Concern#9: The measuring instruments used are PSNR, MSE, RMSE, SSIM and NCC. It is only used for imperceptibility measurement only. Not really explained how to measure it. Then for NCC, is it used to calculate the quality of the stego image or the extraction result?

Author response: We are sorry, and thank you for the comments. Yes, it is used for imperceptibility measurement of the image to analyse the image's quality on a different angle, and NCC is used to check whether the stego and cover image are identical.

 Author action: we updated the manuscript and added the descriptive explanation of quality assessment metrics and their formulas to measure these QAM’s for Cover and Steg-images. Presented in detail in sections 4.1, 4.5, and 4.6, respectively (highlighted in the revised manuscript)

Reviewer#1, Concern#10: It can be seen that the PSNR measurement results are very good, but this value is very strange. How is it possible for an image with a larger dimension to produce a worse PSNR value? In fact, if the image/cover is bigger, the PSNR will be better because the message size is not significant compared to the image size.

Author response: Thanks for the keen comments, Yes, you are right about how it is possible for an image with a larger dimension to produce a worse PSNR value.

The original images obtained for testing were of dimensions 128 * 128, but to check for higher dimensions, I stretched them in Paint. Therefore, the quality of the image was degraded, and when the text was encoded in it, it further degraded the quality; hence, the PSNR dropped instead of increasing. Your valuable comment led me to the conclusion that I have incorporated absolute PSNR. Now, I have modified the values to relative PSNR (to the image's original quality), and it rectified your claim, as provided in the same table.

Author action: We updated the manuscript by modifying all experimental results of all Tables presented in the experimental and result section.

Reviewer#1, Concern#11:  site this relevant paper 

Sadad, Tariq, et al. "A Review on Multi-organ Cancer Detection Using Advanced Machine Learning Techniques." Current medical imaging 17.6 (2021): 686-694.

Author response: Thanks to the anonymous reviewer for the comment. We added the excellent and updated research work mentioned by the reviewer. We also cite the papers and add these in references. 

Author action: We updated the manuscript accordingly by adding and highlighting reference 17.

Reviewer 2 Report

This paper presents a steganography algorithm for colour images. The algorithm was investigated with the images and the results are presented. There are some main negative aspects in the manuscript:

1.The abstract should be modified and more refined.

2. The algorithm should be more clearly stated.

3. There are many typos and grammatical errors. In some places, it is difficult to understand. Need an extensive editing and writing style. and need a complete proof reading.

4. In literature review, it would be better to put the paragraph (210-223) in a table format.

5. Rewrite the paragraph (226-234)

6.  Follow consistency in tense and passive form.

7. Figures 5 and 9 are same.

8. Presentation of the research content should be revised in a comprehensive way

Author Response

Reviewer#2, Concern#1: The abstract should be modified and more refined.

Author response: First of all, thanks for the suggestions. The revised manuscript's abstract has been modified and has been greatly improved. We revised the abstract and redescribed it accordingly.

Author action: We updated the manuscript.

Reviewer#2, Concern#2: The algorithm should be more clearly stated.

Author response: we appreciate this deep and constructive comment and are very thankful to the reviewer for pointing out improvements in the manuscript. We revised the said section and updated the paper accordingly. 

First, we elaborate on the notations and mathematical formulation. Then the step-by-step procedures of the embedding and extraction algorithms are explained, and how the message is embedded within the image and the extraction process of the message is clearly explained in the revised version.

Author action: We revised and updated the manuscript accordingly. In response to the reviewer's valuable comment, we highlight our manuscript on page 14.

Reviewer#2, Concern#3: Many typos and grammatical errors exist. In some places, it is not easy to understand. Need an extensive editing and writing style. and need a complete proof reading.

Author response and action: We are sorry for confusing you due to the lack of sufficient explanations and typos.  We are thankful for your valuable comments on the improvement of this work. We have thoroughly proofread the whole manuscript by an English expert and made extensive corrections in the revised manuscript. 

Reviewer#2, Concern#4: In literature review, it would be better to put the paragraph (210-223) in a table format.

Author response: We are agreed with reviewer concern.

Author action: We have added the details from lines 210-223 in the table, highlighted on page 5, Table 1.

Reviewer#2, Concern#5: Rewrite the paragraph (226-234)

Author response: We are sorry that we did not explain this clearly. We appreciate your question, and thank you for pointing out the main concern.

Author action: We have updated the manuscript by rewriting the relevant description to address the valuable comments of the reviewer and removed the ambiguity in the revised manuscript.

Reviewer#2, Concern#6: Follow consistency in tense and passive form.

Author response: We are sorry that we did not state the paper clearly and made it ambiguous. We appreciate the deep and constructive comments of the reviewer.

Author action: We revised and updated the whole manuscript to remove ambiguity and did our best to make the writing of the manuscript uniform and clear according to the reviewer's comments.

Reviewer#2, Concern#7: Figures 5 and 9 are same.

Author response: We are sorry for confusing readers due to the extra explanation. Thanks for your keen observation and valuable suggestions for the quality improvement of the manuscript. The Figures were duplicated for explanation purposes only. We removed Figure 9 in the revised manuscript and arranged the figure numbering accordingly.

Author action:  We have revised the manuscript to remove ambiguity for addressing the main key point of the reviewer changes highlighted on pages 10-11.

Reviewer#2, Concern#8: Presentation of the research content should be revised in a comprehensive way.

Author response: We have tried our best to present the research contents comprehensively and hope that the highlighted contents in the revised version of the manuscript will be according to the reviewers' standards.

Author action: We revised the whole manuscript and updated it according to the reviewers' valuable comments.

Reviewer 3 Report

The presented multi-perspective-based steganography approach can be a good addition provided the following modifications are incorporated:

1. In the abstract, some more information on the proposed work and results achieved can be presented rather than presenting the introductory content.

2. The keyword Copyright has not been used in the abstract and even hardly used in the manuscripts also. 

3. What is the opinion of the authors about the use of multiple formats, has it any impact on the quality of stego-image, capacity and robustness? I would suggest conducting some tests on various algorithms/techniques using multiple formats.

4. Iin fig.  2, it has been mentioned that authentication is one of the facets/features, how? Steganographic techniques are no way can achieve authentication.

5. I would suggest presenting the numerical illustration step-by-step and flow diagram.

6. In fig. 10, p1,p2,p3 contain all same formats, then, how does they are different perspectives?

7. The considered existing techniques are old. I suggest to follow some recent works with their merits and issues

 7a. https://doi.org/10.1049/cit2.12130

 7.b. https://doi.org/10.1007/s11042-022-13630-4

8. Revise the conclusion and include future directions and implications

Author Response

Reviewer#3, Concern#1: In the abstract, more information on the proposed work and results can be presented rather than the introductory content.

Author response: First of all, thanks for the reviewer's suggestions. With the help of the reviewer's comments, we feel that the revised manuscript has been greatly improved. We revise the abstract according to the reviewer’s comments.

Author action: We updated the abstract accordingly.

Reviewer#3, Concern#2: The keyword Copyright has not been used in the abstract and even hardly used in the manuscripts also. 

Author response: We are sorry for the inconvenience. Thanks for the keen observation of the reviewers. The “copyright” keyword has been removed in the revised manuscript.

Author action: we delete the copyright keyword.

Reviewer#3, Concern#3: What is the opinion of the authors about the use of multiple formats, has it any impact on the quality of stego-image, capacity and robustness? I would suggest conducting some tests on various algorithms/techniques using multiple formats.

Author response: We are sorry for confusing you due to the lack of sufficient explanation about the basic criteria of image steganography, such as capacity, robustness, quality of stego-image etc. We appreciate the valuable comment. Yes, it impacts the said basic parameters, which is why we use multiple formats of different dimensions. Therefore, we used different formats (png, BMP, jpg, tiff etc.) images of distinct dimensions such as 128x128, 256x256, 512x 512, and 1024x1024 for critical analysis on various perspectives to find the impacts of these formats. Our basic objective from this is “that which format image is better for embedding a secret message in terms of reliability”.

We had already conducted tests, as shown in Table 4 to Table 7. The details of some tests and comparative analyses are presented in Table 8 of the revised manuscript.

Author action: We updated the manuscript by adding a descriptive explanation in the paper to address the valuable comment of the reviewer.

Reviewer#3, Concern#4: Line figure 2, it has been mentioned that authentication is one of the facets/features, how? Steganographic techniques are no way can achieve authentication.

Author response: We are sorry, and thank you for pointing out the missing explanation for the future hot research topic. You are right that straightforward authentication cannot be achieved through steganography. However, it is used in general terms for sender (Encryption) and receiver (Decryption). In this paper, we have no concern (work) on authentication however, it can be used as a steganography feature. For authentication, some extra data may be added at the sender side through which the receiver can authenticate. For further detail, the specific algorithm Intelligent Identity Authenticator (IIA) is used in steganography at the sender side.

Author action: No action is taken against this comment.

Reviewer#3, Concern#5: I would suggest presenting the numerical illustration step-by-step and flow diagram.

Author response: We are sorry, and thank you for pointing out the missing explanation. The Suggested comment will greatly improve the paper. We added the step-by-step procedure explanation and pseudo code of the embedding, extraction, and MLEA in algorithms 1, 2, 3 etc. We also explain the mathematical formulation of the proposed algorithms using six functions as α, β, γ, ?, Ω, φ presented in section 3.3. We updated the said section by adding some more explanations and flow diagrams to address the reviewer's valuable suggestion.

Author action: We updated the manuscript by adding the descriptive explanation and flow diagram. The reply is highlighted in Figure 11.

Reviewer#3, Concern#6: In fig. 10, p1, p2, p3 contain all same formats, then, how does they are different perspectives?

Author response: Thank you for giving us a chance to clarify this statement. Actually, we have used four different perspectives (P1, P2, P3, and P4) for different formats of images (i.e. png, jpg, tiff, BMP, etc.). Following are the different perspectives:

  • In P1, we used these different formats (png, bmp, jpg, tiff etc.) images with the same dimension 512x 512, and size of the message 12 kb.
  • In P2 we also used these distinct formats image with different dimensions 128x128, 256x256, 512x 512, 1024x1024, and secret message size is 12 kb
  • P3 present different formats of image with the same dimension, but the message size is different such as 6, 8, 10, and 12 kb’s

The main purpose of using these different perspectives (p1-p4) on these different formats of image (png, bmp, jpg, tiff etc.)  is to critically analyze the proposed algorithm for finding the best and appropriate procedure. The reason of these different perspectives is to find which formats of the image are better and for which size of the message and also to identify how many amounts of the secret message can be embedded in which dimensions of an image robustly and perceptibly to fulfill the basics needs of the image steganography.

Author action: We have highlighted the relevant text for reviewer understandability.

Reviewer#3, Concern#7: The considered existing techniques are old. I suggest to follow some recent works with their merits and issues.

 7a. https://doi.org/10.1049/cit2.12130

 7.b. https://doi.org/10.1007/s11042-022-13630-4

Author response: Thanks to the reviewer for the comment. We have cited the papers and added these in references.  In the revised manuscript, we have compared some relevant and recent methods to show the significance of our method using different perspectives.

Author action: We have updated the manuscript accordingly.

Reviewer#3, Concern#8: Revise the conclusion and include future directions and implications.

Author response: Thanks to the reviewer for suggesting this concern. We have revised the conclusion of the manuscript and added future directions and implications.

Author action: We have updated the manuscript accordingly.

Round 2

Reviewer 2 Report

Table 1 can have column headers.

Manuscript can go with miner error corrections-typos and punctuations.

Author Response

Reviewer#2, Concern#1: Table 1 can have column headers.

Author response: Thank you for the correction. We are agreed with reviewer concern.

Author action: We have updated the manuscript accordingly and added the headers to Table 1.

Reviewer#2, Concern#2: Manuscript can go with miner error corrections-typos and punctuations.

Author response:  Thank you for your valuable comment.

Author action: We have revised the whole manuscript and corrected many typos and punctuations.

Reviewer 3 Report

The paper has been revised sucessfully

Author Response

Thank you for your time and fruitful suggestions.

Regards,